# On the Pathogenicity of the Oral Biofilm: A Critical Review from a Biological, Evolutionary, and Nutritional Point of View

**DOI:** 10.3390/nu14102174

**Published:** 2022-05-23

**Authors:** Johan Peter Woelber, Ali Al-Ahmad, Kurt Werner Alt

**Affiliations:** 1Department of Operative Dentistry and Periodontology, Faculty of Medicine, University of Freiburg, Hugstetter Str. 55, 79106 Freiburg, Germany; ali.al-ahmad@uniklinik-freiburg.de; 2Center of Natural and Cultural Human History, Danube Private University, Steiner Landstrasse 124, 3500 Krems-Stein, Austria; kurt.alt@dp-uni.ac.at

**Keywords:** plaque, biofilm, caries, gingivitis, periodontitis, evolution, lifestyle, diet, nutrition

## Abstract

Plaque control is one of the most recommended approaches in the prevention and therapy of caries and periodontal diseases. However, although most individuals in industrialized countries already perform daily oral hygiene, caries and periodontal diseases still are the most common diseases of mankind. This raises the question of whether plaque control is really a causative and effective approach to the prevention of these diseases. From an evolutionary, biological, and nutritional perspective, dental biofilms have to be considered a natural phenomenon, whereas several changes in human lifestyle factors during modern evolution are not “natural”. These lifestyle factors include the modern “Western diet” (rich in sugar and saturated fats and low in micronutrients), smoking, sedentary behavior, and continuous stress. This review hypothesizes that not plaque itself but rather these modern, unnatural lifestyle factors are the real causes of the high prevalence of caries and periodontal diseases besides several other non-communicable diseases. Accordingly, applying evolutionary and lifestyle medicine in dentistry would offer a causative approach against oral and common diseases, which would not be possible with oral hygiene approaches used on their own.

## 1. Introduction

At present, plaque control is considered the most important preventive factor concerning caries, gingivitis, and periodontitis [1,2,3], and impressive technological developments such as the use of sonic or ultrasonic electric toothbrushes have been made to further reduce the amount of plaque around the teeth [4,5]. This preventive “knowledge” of oral hygiene is highly prevalent, and several national health surveys showed that, in industrial countries, more than 90% of the population regularly brushes their teeth once or twice daily [6,7,8]. However, although oral hygiene is performed in this high prevalence, industrial countries especially are confronted with tremendously high rates of caries and periodontal diseases [7,9,10]. This imbalance between preventive measures and the occurrence of oral diseases accordingly questions the efficacy and causality of the approach of plaque control. Furthermore, it raises the question of why *Homo sapiens* should be the only species on earth whose oral health is dependent on a cultural behavior such as oral hygiene.

The idea that plaque is a causative factor for the development of oral diseases has a long history in human cultural evolution. The first descriptions of rudimentary oral hygiene efforts can be found in Mesopotamia (3000 BC), China (1500 BC), and Egypt (before 1500 BC) [11]. Early cultures from Egypt, Greece, and Italy provided descriptions of oral health behaviors and dentifrices, as recommended by Hippocrates (460–377 BC) [11,12]. A modern breakthrough was achieved by Antoni van Leeuwenhoek (1632–1723 AD), one of the inventors of the first microscope, who described plaque as an animated substance [13]. Van Leeuwenhoek also associated plaque with gingival inflammation and recommended oral hygiene procedures.

The most influential modern studies that connected plaque accumulation to gingival inflammation and caries were performed in the 1960s in Scandinavia. Among these, in an “experimental gingivitis” study, Löe et al. showed that uncontrolled plaque accumulation for 14 days led to a significant increase in gingival inflammation and that oral hygiene was able to reduce plaque and, accordingly, the gingival inflammation [14]. Besides the small number of participants (*n* = 12), there was also no further background information concerning lifestyle factors such as diet (“standard Scandinavian diet”), smoking, physical exercise, weight, or stress. However, this study seemed to clearly show the strong relationship between plaque and gingivitis.

Lindhe et al. validated these principles for the development of periodontitis in an animal model of beagle dogs, with an interventional group with regular tooth cleaning and a control group without oral hygiene for 4 years [15]. Due to the consistent health in the interventional group and the periodontal destruction in the non-hygiene group, the authors concluded that oral health is clearly dependent on meticulous oral hygiene, even though the idea of the necessity of oral hygiene in other species (like beagle dogs) might sound unfamiliar since oral hygiene is usually not found in wild animals. Nonetheless, this cause relationship seemed convincing. However, there was no further critical discussion about the highly untypical diet of these beagle dogs except for a reference to a “soft pellet diet” [16]. Hamp et al. described the beagle study diet as “a soft sticky diet consisting of dog sausage, potato flour, sugar and water” [17], which represents an obviously non-natural, species-inappropriate diet.

With regard to caries, von der Fehr et al. evaluated the principles of the experimental gingivitis by Löe et al. in a controlled study on 12 participants who were either allocated to an experimental group without oral hygiene and additional sugar intake for 23 days or a control group without oral hygiene and without additional sugar intake [18]. Even though the sugar-supplemented group developed carious lesions more rapidly, the control group also showed an increase in the caries index. Accordingly, the researchers concluded that proper oral hygiene with additional fluorides is necessary for maintaining oral health. The problem of the “background“ diet of the control group, which also included sugary foods, was not part of the broader discussion. 

Apart from the discussion concerning whether plaque control is rather symptomatic or causal, the following decades proved the efficacy of oral hygiene with regard to periodontal diseases [1,3], while the evidence concerning caries was only significant with the additional use of fluorides [19,20], as plaque reduction without fluorides was shown to be ineffective against caries [20].

However, even if there is an undebatable preventive effect of oral hygiene on gingivitis and possible caries, the implicit “final aim” of this kind of prevention would be plaque-free conditions in every individual. This aim seems as unrealistic as uncausal, as proven by the discrepancy between the already high level of oral hygiene efforts of industrialized populations and the still remaining high prevalence of oral diseases. Furthermore, if the causes of these diseases may lie in other factors than plaque, plaque control would accordingly only be a symptomatic approach that would blur the symptoms but not the underlying causal factor [21,22]. This implies at least two harmful or detrimental consequences for personal and even public health care: the primary bodily “warning signal” in the form of gingivitis, e.g., as a consequence of excessive sugar consumption or micronutrient deficiency, would not lead to a cause-related therapy (sugar cessation or supply of micronutrients) but to a symptomatic plaque reduction [21,22,23]. While the gingivitis would get better, the malnutrition would not and may lead to further diseases (e.g., overweight or retinal hemorrhaging). On the other side, looking from a public health perspective, a cause-related therapy as a first step would save immense financial resources due to the prevention of other subsequent non-communicable diseases (NCDs), which is also described in the common risk factor approach [24,25].

Accordingly, this review aims to critically question the assumption that plaque alone is the cause of most oral diseases in the form of caries and periodontal diseases by means of the following hypotheses derived from modern biology, evolutionary medicine, and nutritional science:I.The formation of biofilms (such as dental plaque) is a normal process in nature and applies to all species where hard surfaces are in contact with fluids (such as water or saliva).II.Since bacterial biofilms were already colonizing planet earth long before the successful coevolution of animals and humans, oral health must principally be possible even in the presence of biofilms.III.In human evolution, the principal process of biofilm formation has not changed, but rather, several harmful lifestyle factors in whole societies, such as poor diet, sedentary behavior, overweight, smoking, and chronic stress, have become prevalent.

Based on these factors, the review will discuss possible cause-related approaches for oral health under modern circumstances.

## 2. Evolution of Biofilms in Nature, Animals, and Humans

A great deal of evidence exists to suggest the presence of a microbial biosphere in terrestrial habitats for 3.2 billion years and in marine settings for at least 3.42 billion years [26]. This highlights a long period of evolution that led to a high morphological and metabolic diversity of bacteria, which enables them to colonize most biotic and abiotic habitats on Earth. Until the 1970s, microbiologists mainly focused their research on free planktonic microbial cells and did not actively investigate the ability of bacteria to form a biofilm [27]. Costerton et al. [28] were the first researchers who stated that most bacteria live in biofilms, after which many other reviews confirmed the role of biofilms as the main form of bacterial occurrence in microbial habitats [27,29,30]. Flemming and Wuertz [27] calculated a total number of 1.2 × 10^30^ cells of bacteria and archaea on Earth and defined their five biggest habitats: deep oceanic subsurface (4 × 10^29^), upper oceanic sediment (5 × 10^28^), deep continental subsurface (3 × 10^29^), soil (3 × 10^29^), and oceans (1 × 10^29^). Furthermore, the authors depicted the total number of bacterial cells in different eukaryotic habitats, including the human gut (4 × 10^23^ cells), dental plaque (8 × 10^21^ cells), and skin (1 × 10^21^ cells). Overall, Flemming and Wuertz assumed that 40–80% of microbial cells on Earth exist as biofilm [27]. The large numbers of microbial cells described above also serve to highlight the significant taxonomic and subsequent metabolic diversity of biofilm residents, thereby emphasizing the fact that natural biofilm such as dental plaque has a high capability of adaptation to environmental changes such as nutrition. Costerton et al. [28] defined biofilms as “matrix-enclosed bacterial populations adherent to each other and/or to surfaces or interfaces”. It should be emphasized that microorganisms within the biofilm are embedded in a self-produced matrix of extracellular polymeric substances (EPS), which itself consists of heterogeneous substances that mainly include polysaccharides, proteins, and extracellular DNA [27]. In the biofilm mode of life, microorganisms are capable of developing other emergent properties that cannot be found in free-living cells [27,30], including enhanced tolerance toward antimicrobials, such as biocides, antibiotics, and disinfectants. One interesting mechanism that enhances such emergent properties is the horizontal gene transfer, which enables the acquirement of new genes among biofilm cells [27,31,32]. The main mechanisms of gene transfer are transformation, transduction, and conjugation. From an evolutionary point of view, the above-mentioned features of biofilms facilitate an adaptation of biofilm bacteria to different environmental factors and could be exploited to modulate the biofilm function, i.e., through varying nutrients.

The human oral cavity comprises different niches that harbor the second-most abundant microbiota after the gastrointestinal tract, which includes 772 prokaryotic species as contained in the expanded Human Oral Microbiome Database (eHOMD) [33]. Not only does the species diversity as such promote the metabolic diversity of the oral biofilm but also the capability of single species such as *Streptococcus mutans* to change their metabolic activity (i.e., respiration and fermentation) [34] depending on the environmental factors so that nutrition, oxygen, and pH-value increases the adaptation capacity of oral biofilm. Besides human saliva, the oral biofilm depicts the main mode of life for oral bacteria. It should also be mentioned that salivary bacteria do not only consist of free-living cells but also of biofilm aggregates as swimming biofilm flocks, which renders the term “planktonic oral bacteria” inappropriate [35,36,37].

This evolutionary omnipresence of biofilms and its consequences can be studied when adherent oral biofilms—mainly from the teeth—are removed, and a clean tooth surface immediately provides a new habitat for biofilm formation. Due to the ideal environmental factors within the oral cavity, including a great diversity of salivary components, such as proteins, ions, and enzymes, salivary pellicles are quickly built, on which the microbial adhesion and biofilm maturation take place [38]. Besides the great diversity of microbiota in oral biofilm, the diversity and function of EPS should be also considered for the development of multitargeted strategies to eradicate oral biofilm [39]. In this context, the main EPS functions, including mechanical stability, scaffolding, adhesion–cohesion, and protection against adverse environmental conditions, should be considered, and the term matrixome cannot be separated from the term microbiome for a mutualistic and integral view of oral biofilm and its related oral diseases [39]. Xiao et al. [40] showed the virulence and role of EPS in caries lesion development. The authors visualized complex 3D architecture consisting of highly compartmentalized and mechanically stabilized acidic and EPS-rich microenvironments throughout an intact mixed-species biofilm. These microenvironments consist of densely EPS-packed bacterial islets on the tooth sur-face and shelter the bacteria against salivary buffering effects. Hence, they allow pH niches, promoting acid-tolerant bacteria and leading to caries lesions. These heterogeneous layers of EPS consist of soluble and insoluble glycan and were derived from glycosyltransferases in the presence of sugar, which means that sugar is an enhancement factor for a highly stable biofilm structure. The role of frequent sucrose consumption and the dietary composition in general for microbiota fluctuation and modulation of the oral biofilm was recently shown in a long-term clinical study [41,42]. Hence, sugar enhances the cariogenic capability of oral microorganisms both by altering the microbiota and by enhancing the mechanical stability of the oral biofilm.

The necessity of such a comprehensive view on caries development is highlighted by several key articles on the subject. Rosier et al. [43] outlined the history of various hypotheses to explain the relationship between oral biofilm and disease, which began in the 19th century with the non-specific plaque hypothesis that considered the total plaque as an etiological factor for caries. In the 20th century, the specific plaque hypothesis evolved by depicting the activity of certain species involved in the disease. Once again, the non-specific plaque hypothesis was updated by considering the activity of the total biofilm microbiota. Marsh [44] coined the term “ecological plaque hypothesis”, which proposes that caries are formed as a result of a shift in the balance of microorganisms in the plaque towards disease-related species. Takahashi and Nyvad [45] extended the ecological plaque hypothesis by explaining caries as the outcome of different stages, including a high level of diversity of oral microorganisms that are characteristic of dynamic stability and the acidogenic and aciduric stages of caries. Significantly, none of these hypotheses considered the above-mentioned complicated interactions of EPS (matrixome) with biofilm microbiota and its role in disease development—presumably due to methodological limitations or the absence of a multi-approach strategy. Independently of the hypothesis concerning the caries etiology and causative agents, nutrition was shown to always be a major modulating factor not only over the course of recent short or long-term experiments performed in vitro or in vivo but also throughout human evolution [46].

In today’s perception, it seems indisputable that oral hygiene using a toothbrush is required to maintain oral health and prevent caries, gingivitis, and periodontitis. However, the question remains whether the microbial and biochemical properties of oral biofilm have been sufficiently exploited to enhance the protective outcome of the widely-used approaches for maintaining oral hygiene. The fact that the prevalence of the aforementioned oral diseases is still high despite the intensive use of diverse oral hygiene techniques indicates that this is not the case, and studying the oral microbiome of wild and captive animals may help to find a conclusive answer to this question. A recent 16S rRNA high-throughput sequencing study revealed that captive long-tailed macaques had distinct oral–gut microbiota profiles and lower bacterial richness compared to those observed in wild macaques [47]. The authors suggested that there is a significant effect of the wild environment and especially the diet on the oral microbiome of these animals. A recent study compared the microbiome of human saliva with that of captive great apes, including bonobo, chimpanzee, gorilla, and orangutan [48]. The authors found that the highest abundance of caries-associated bacteria was detected in human saliva. Furthermore, periodontitis-associated bacteria were more abundant in human saliva than in all other tested species of great apes. The authors suggested that lifestyle factors, especially the diet, are the trigger for the higher risk of humans for caries and periodontitis. However, these findings in the aforementioned study should be confirmed by analyzing and comparing the microbiome of oral biofilm samples.

Only a few studies have been conducted on the ancient dental calculus of animals. Using shotgun sequencing of dental calculus in ancient Egyptian baboons held in captivity during the late Pharaonic era (9th–6th centuries BC) and of two historical baboons from a zoo, Ottoni et al. [49] delivered some interesting indications concerning the influence of captivity on the oral microbiome of these animals. The authors did not exclude the influence of habitations such as diet on the microbiome of the animals.

## 3. Human Evolution, Nutrition, and the Oral Microbial Colonization

About 250 mya, mammals evolved out of reptiles (synapsids), and about 65 mya, the first primates evolved in the Eocene. About 8 mya, the evolutionary line of humans separated from that of today’s apes, continuing in the evolution of the hominins about 3 mya. According to the latest fossils, our species *Homo sapiens* is now dated to 300,000 years ago [50,51]. Based on the aforementioned 3.2 billion years of bacterial pre-colonization, it seems logical that even the first mammals, primates, and *Homo sapiens* evolved in a bacterial environment and were also colonized by bacteria themselves. Based on this, it does not seem surprising that modern microbiology and immunology increasingly underline the importance of commensal and symbiotic bacteria for overall health and a common view on host–microbe interactions [52,53].

The process of oral bacterial colonization starts shortly after birth and is largely based on the surrounding microbial environment (mainly of the mother). This is underlined by a study by Holgerson et al. [54] showing a reduced oral taxa richness of infants delivered by Caesarian section compared to vaginal delivery, even 3 months after birth. For the longest time in its history, this bacterial environment was based on nature. Until then, the most important factors influencing the microbiome were primarily nutrition and genetic conditions [55,56]. With regard to nutrition, the availability of plants and animals in the immediate and wider environment and the presence of water ensured survival [57]. Before the transition to agriculture, nutrition was based on the gathering of (predominantly) plants (80%) and on hunting wild animals and fish (20%). The fibrous nature of these foods, which were moderately processed, led to abrasion of the teeth (Alt et al., 2022, submitted). Furthermore, the natural environment of hunting and gathering both led to a constant high level of physical activity and periods of food shortage and fasting. Fertility restrictions induced by this kept population densities low for a long time. The diet of prehistoric man was determined by the seasons, seasonally available resources, climatic conditions, and the respective biotope. A strongly community-focused solidarity principle determined the subsistence strategies of small socio-economic groups [58].

The prevalence of oral diseases such as caries and periodontitis was quite low in our Pleistocene ancestors [59,60,61,62,63]. Even if there are few reports of carious teeth from *Homo sapiens* in the Pleistocene [64], these can be seen as an early transition to the processing of carbohydrate-rich foods (like acorns). Furthermore, these findings have to be seen critically since not every defect in a tooth can be causally attributed to caries, and excessive attrition can also lead to defects reaching out to the dental pulp, resulting in endodontic diseases.

## 4. Neolithization and Industrial Revolution, Cultural Evolution, and the Role of Plaque in a Dramatic Change in the Nutritional Environment

The history of *Homo sapiens* experienced two major revolutions in the form of Neolithization (over 12,000 years ago) and the Industrial Revolution (beginning in the 19th century), each of which had drastic consequences for life, diet, and the level of physical activity and (oral) health.

The Neolithization was characterized by a change from hunting and gathering to a sedentary lifestyle based on the cultivation of cereals and animal husbandry. It is assumed that this process was initiated during a warm period and favorable climatic conditions in the Levant in 12,000 BC. The Neolithization was also the starting point for an immense cultural evolution with changing values in communities and resource distribution [58]. In Central Europe, peasant settlement occurred in the course of the 6th millennium through the infiltration of foreign population groups from the Carpathian Basin [65,66]. The essential role of assimilation in the Neolithization of Europe is confirmed by data on population genetics based on the analysis of ancient DNA (aDNA) [67,68].

How successfully *Homo sapiens* has acted since his emergence is shown, among other things, by the fact that he has managed to successfully adapt to his environment globally, which was facilitated by cultural evolution [69,70]. The core factors that have ensured survival in the course of human history include not only food and drink but also cultural achievements such as clothing, housing, energy use, etc. These achievements cannot be defined without cultural and social standards [71]. Furthermore, human beings themselves are neither conceivable nor capable of survival without social integration.

With regard to nutrition, *Homo sapiens* showed remarkable flexibility to different dietary environments, which allowed settlement of even distant eco-systems. As an example, the Inuit are able to achieve stable communities purely based on animal proteins, while other populations such as those in the Andes are primarily living on plant foods. Accordingly, it does not seem possible to speak of a single “natural” way of eating for humans.

However, an important difference that goes beyond this natural variation of a wholesome diet based on different ecosystems is the processing of food after the Industrial Revolution. While the processing of foods in the form of cooking is considered as being an important step in evolution and brain development due to the higher availability of nutrients compared to uncooked foods [72], the main forms of “processing” today describes the extraction of fibers and antioxidants in favor of pure sugar, starch, and fats [21,73,74]. The consumption of processed or ultra-processed foods is one of the main characteristics of the so-called “Western diet”, which is the term used to describe today’s average diet in Western industrial countries [75]. Furthermore, it includes the use of food additives such as salt, other flavor enhancers, and preservatives. The detrimental health effects of the Western diet are quite well-researched and consistently described as pro-inflammatory, dysbiotic, hypercaloric, disease-promoting, and habituation-forming by both detrimental effects on metabolism and brain neurotransmission (such as via dopamine) [75,76,77,78,79]. While for the longest time in human evolution, it was life-threatening to not find food, the modern dietary environment has become so harmful that diet is considered to be the most significant contributing behavioral factor to premature death [80]. The logical consequence of this knowledge would be to change the diet and avoid these ultra-processed foods. However, besides the intrinsic rewarding effects of processed foods (such as sugar), the complete social process of nutrition in modern societies is also significantly influenced by modern economic and social systems such as capitalism and social rituals. Within the social rituals, processed foods containing large amounts of sugar were embedded in a variety of recurring habits such as sugared coffee, sugared cakes in the afternoon, and sugary foods on birthdays, Easter, Halloween, and Christmas, as only some examples [81,82]. Within the capitalized system, food industries influence “consumers” through advertisements and political lobbyism to continue a Western type of diet [83,84,85]. Even in the field of dental research, the sugar industry influenced the research agenda of the U.S. National Institute of Dental Research (NIDR) to avoid research dealing with sugar reduction strategies [86]. Not surprisingly, the sugar consumption (as only one aspect of the Western diet) rose from below 1 kg per capita per year in the 17th century to about 30–40 kg in modern industrialized countries [87].

These dramatic changes in diet were also accompanied by several “new” risk factors such as smoking, increasing psychological stress, decreased physical inactivity, and antibacterial strategies (such as disinfection and the use of antibiotics) resulting in a loss of important commensal bacteria for immune system development (like *Bifidobacterium* spp., *Prevotella* spp., *Xylanibacter* spp., *Faecalibacter* spp.) and the increase of autoimmune diseases and allergies [80,88,89,90,91].

It seems unsurprising that these changes also had a severe impact on the oral microbiome and dental plaque. Adler et al. were able to display this transformation (from pre-Neolithic times to the Neolithic and Industrial Revolution) by microbiological analysis of calcified plaque from different points in time of human evolution [46]. They were able to show a dramatic increase of caries and periodontitis-related strains (*S. mutans* and *P. gingivalis*) from hunter-gatherer to modern times. This is in line with the enormously high prevalence of caries and periodontitis in modern industrialized countries. Furthermore, the data demonstrated a shift from a highly diverse biofilm with proportions of the phylum *Firmicutes* and unclassified bacteria in hunter-gatherer times to a less diverse biofilm with high proportions of the phylum *Proteobacteria*.

## 5. Symptomatic and Causal Approaches against Caries and Periodontitis and Evolutionary and Lifestyle Dentistry

Summarizing the presented evidence, it is obvious that the natural process of plaque forming in the oral cavity has not changed between hunter-gatherer and modern times but rather that the surrounding environment is characterized by novel disease-promoting risk factors (Figure 1) [92].

It can only be speculated why researchers primarily investigated symptomatic treatments and preventive concepts against caries and periodontitis such as the use of fluorides and oral hygiene. Certainly, the presented intrinsic and extrinsic motivation to continue a Western-style diet and the introduction of (historically seen) rather “new” risk factors, which were either addictive (like smoking), too elaborate (like physical activity), or contradictory to health changes (like chronic stress), played a major role [90,91,93,94]. Although the investigated symptomatic treatments such as fluoridation and plaque control showed a certain degree of efficacy, they were also shown to be unable to fully prevent oral diseases in the presence of persisting risk factors: A recent systematic review showed that oral plaque control on its own (without the use of fluorides) was not able to prevent caries [20]. Bernabe et al. [95] showed that although fluorides were able to lower the incidence of caries with increasing sugar consumption, they were not able to stop the incidence. Bergström and Eliasson demonstrated that even high standards of oral plaque control were not able to stop periodontitis-associated bone loss in smokers [96]. Accordingly, even when plaque control is performed by most people in industrial countries, the persistence of these risk factors still leads to the high global burden of caries and periodontitis, which together remain the most common diseases of mankind [9,10]. Worse than that, these factors led to the current pandemic of non-communicable diseases, which is the main reason for premature death and disability-adjusted life years (DALYs) [80].

Based on this evidence, a causal approach against caries and periodontitis must accordingly address the (individual) risk factors, which today differ from a “natural” environment seen in hunter-gatherer times. This claim neither excludes current plaque control efforts nor a life under modern circumstances. It rather prioritizes the therapeutical and preventive approaches required to maximize the overall health of individuals. This approach is also described as the common risk factor approach (CRFA) [25]. As an example, while investing counseling time and health education on plaque control (the outcome of which would be beneficial for periodontitis and maybe caries), the same amount of time and efforts invested in sugar cessation would both be beneficial for periodontitis, caries, obesity, diabetes mellitus type II, coronary heart disease, and premature death at the same time. The same applies to smoking, physical activity, and stress reduction. Interestingly, all these “upstream” factors of the CRFA are in line with the health behavior in hunter-gatherer times. Applying the principles of evolutionary medicine [97], the aim of evolutionary or lifestyle dentistry would in the first line be to identify and modify evolutionary risk factors and in the second line to add symptomatic approaches such as fluorides and plaque control. Considering a large number of dental teams in industrial countries, the tremendous treatment costs and DALYs of only sugar-associated diseases, with 4.1 million DALYs and USD 172 billion per year worldwide [98], one can imagine the potential of such an approach. It would be extremely beneficial on both an individual and public health basis.

Distinguishing the existing evidence on caries and periodontitis prevention with regard to evolutionary dentistry, there is already significant evidence for the efficacy of causal therapies:-Dietary patterns focusing on a higher intake of non-processed (fibrous and antioxidant-rich) and an avoidance of processed foods were consistently shown to reduce gingival inflammation—even despite constant or higher plaque values [99,100,101,102]. Whole-food diets were shown to reduce periodontal pathogens without any additional mechanical therapy [103,104]. Interventional trials showed that avoiding sugar can significantly reduce the amount of *S. mutans* [105,106]. Low-sugar diet patterns are significantly associated with a lower caries experience [107].-Smoking cessation can stop further smoking-related periodontal destruction [108], and not starting smoking is one of the most effective measures to prevent the initiation of periodontal diseases [109].-Increased physical activity was shown to reduce periodontal inflammation [110,111] although unnaturally high levels of physical activity (as seen in high intensive endurance sports) may provoke more periodontal inflammation [112].-Stress reduction (e.g., via mediation or yoga) was shown to significantly decrease plaque and periodontal inflammation [113,114,115].

Of course, the body of evidence for evolutionary dentistry is rather small compared to the studies investigating oral hygiene and fluorides. This discrepancy could potentially be explained in light of the question of which approaches might find more industrial funding. However, this also harbors the chance for public health policy to guide public third-party funding in a cause-related direction. With healthcare spending on the rise, this policy will become more urgent in the future.

## 6. Conclusions

In the light of bacterial and human coevolution, the pathogenicity of plaque itself did not change but rather the modern, unnatural life-style factors leading to both oral and several other non-communicable diseases. Accordingly, the current primary focus on plaque reduction in modern dentistry has to be judged as a rather symptomatic than causal prevention of caries and periodontal diseases. Thus, an evolutionary dentistry approach would primarily focus on disease-promoting risk factors such as Western diet patterns, smoking, sedentary behavior, and chronic stress that evolved during cultural evolution.

## Figures and Tables

**Figure 1 nutrients-14-02174-f001:**
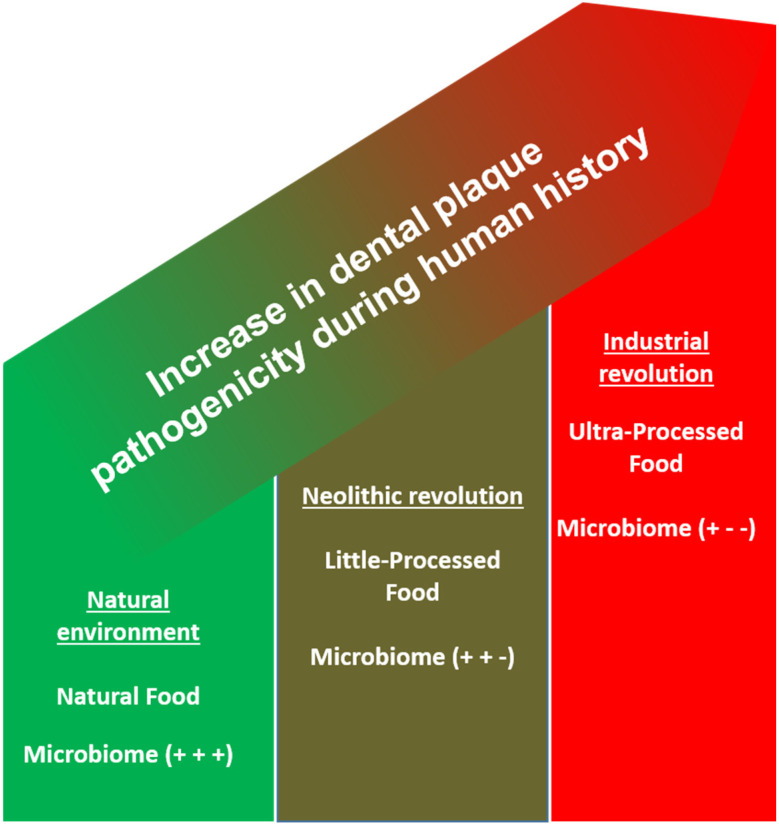
Scheme for the increase in plaque pathogenicity based on the drastic changes in human history. +, health-promoting; - disease-promoting.

## Data Availability

Not applicable.

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
