# Peer review of "On the Pathogenicity of the Oral Biofilm: A Critical Review from a Biological, Evolutionary, and Nutritional Point of View"

_nutrients, 2022, doi:10.3390/nu14102174_

Round 1

Reviewer 1 Report

This is an interesting review. I only have one suggestion.

The authors can make a table to list the process from research article.

Author Response

Dear reviewer, thank you so much for your comment, your efforts and time. We were discussing about your comment as our review was not a systematic review and we thus cannot deliver a flowchart of literature processing. If we got you wrong in this interpretation, we would be glad if you could specify your suggested table. 

Reviewer 2 Report

The manuscript by Woelber, Al-Ahmad, and Alt was a relatively comprehensive research review on the potential roles of oral biofilm in the human patients. The authors seemed to focus on the biological mechanisms and the use for potential treatment targets. There were some moderate concerns:

  • Page 4 Line 175 was unclear. Mechanisms regarding EPS and microenvironment should be added.
  • The type of biofilm and differences should be summarized, such as for Page 6 Section 3 Lines 258-264.
  • References for Section 4 Lines 280-286 need to be added. Some are too descriptive.
  • Consider summarizing the related species for Page 7 Lines 322-326.
  • The conclusion was not evidence-based.

Author Response

Dear reviewer, thank you so much for your constructive comments and your efforts. Following, we want to response to your comments point to point.

1) Page 4 Line 175 was unclear. Mechanisms regarding EPS and microenvironment should be added.

Answer: We added further information about the mechanisms to the text now. (p4l178)

2) The type of biofilm and differences should be summarized, such as for Page 6 Section 3 Lines 258-264.

Answer: We added a corresponding summary at the end of section 4 now. ("Furthermore, the data demonstrated a shift from a high diverse biofilm with quite proportions of the phylum Firmicutes and unclassified bacteria in hunter-gatherer times to a less diverse biofilm with high proportions of the phylum Proteobacteria.") P7L338f.

3) References for Section 4 Lines 280-286 need to be added. Some are too descriptive.

Answer: Thank you for this remark. We added references for this paragraph now. P6L283f.

4) Consider summarizing the related species for Page 7 Lines 322-326.

Answer: Thank you. We added the most influenced species now. P7L328

5) The conclusion was not evidence-based.

Answer: We rephrased the conclusion according to the evidence presented.